# Reconstructing changes in nitrogen input to the Danube-influenced Black Sea Shelf during the Holocene

**Andreas Neumann**[1], **Justus E. E. van Beusekom**[TS1][1], **Alexander Bratek**[1,2], **Jana Friedrich**[1,4], **Jürgen Möbius**[2], **Tina Sanders**[1], **Hendrik Wolschke**[3], **and Kirstin Dähnke**[1]

[1]Institute of Carbon Cycles, Helmholtz-Zentrum Hereon, Geesthacht, Germany[TS2]
[2]Center for Earth System Research and Sustainability, Institute of Geology, Universität Hamburg, Hamburg, Germany
[3]Institute of Coastal Environmental Chemistry, Helmholtz-Zentrum Hereon, Geesthacht, Germany
[4]Department of Nuclear Sciences and Applications, IAEA Marine Environment Laboratories,
International Atomic Energy Agency, 98000 Monaco, Principality of Monaco

**Correspondence:** Andreas Neumann (andreas.neumann@hereon.de)

**Abstract.** The western Black Sea shelf is particularly sensitive to river-induced eutrophication due to river discharge from the Danube River, and accordingly, eutrophication peaked in the 1980s and 1990s due to human-induced nutrient input. Nutrient input to the western Black Sea shelf and eutrophication decreases since the mid-1990s due to the collapse of eastern European economies after 1989 and ongoing mitigation measures to reduce nutrient emissions. The assessment of nutrient inputs to the Black Sea prior to the 1960s however is hindered by the scarcity of information on earlier Danube nutrient loads. Thus, to define pristine conditions to provide a reference for nutrient reduction targets remains challenging. In this study, we aim to trace modern and historical nitrogen sources to the western Black Sea Shelf during the last $\sim 7000$ years with special focus on the past 200 years, using sedimentary records of TOC, TIC, nitrogen, and $\delta^{15}$N to quantify the share of anthropogenic nitrogen.

Our results demonstrate that climate effects determine the relative contribution of riverine nitrogen and pelagic nitrogen fixation to fuel marine primary production on the NW shelf. This balance is not only controlled by the riverine nutrient load, but also by the freshwater volume itself, which controls the intensity of thermohaline stratification and thereby the timing and intensity of nutrient recycling from the deep basin back into the euphotic surface water. In the cold and dry Sub-Boreal climate pelagic N-fixation dominates over riverine N discharge, while in the warm and wet Atlantic climate riverine N discharge appears as dominant N source to sustain primary production on the NW shelf. Stable nitrogen isotopes further demonstrate the increased deposition of nitrogen from human activities across the shelf and the concomitant changes in deposition rates of organic matter, which can be tracked back to perturbations in the plankton due to the human-induced eutrophication. Finally, our stable isotope data indicate that human-induced eutrophication can be traced back to the 11th century CE and highlight that the Danube nutrient load was not pristine for at least the past 900 years.

## 1 Introduction

The Black Sea is a semi-enclosed sea, which is connected to the Mediterranean through the Bosphorus. The limited inflow of saline mediterranean water through the Bosporus in combination with freshwater discharge by rivers creates a strong thermohaline stratification, which separates the ventilated surface water from oxygen-free, euxinic bottom water, thereby creating the largest anoxic water body on earth. The oxycline between the ventilated surface water and the euxinic deep water promotes substantial rates of N-loss by bacterial denitrification (Fuchsman et al., 2019) or anammox (Kuypers et al., 2007[TS3]) in the water column. However, hydrogen sulfide ($H_2S$) in euxinic environments reduces the degradability of organic matter (Raven et al., 2018; Kok et al., 2000), which also preserves the isotopic signature of nitro-

gen therein. Additionally, Möbius and Dähnke (2015) found that the plankton community of the Danube River Plume efficiently assimilates nitrogen from the water and thereby outcompetes ammonium oxidising and denitrifying bacteria. In consequence, the plankton community efficiently keeps the nitrogen in particulate organic matter until this organic matter is eventually deposited on the shelf sediment close to the Danube Delta, so that water column denitrification is not a significant nitrogen sink in the distant shelf region.

The thermohaline stratification makes the Black Sea susceptible to climate and human pressures. The climatic oscillation in the Black Sea region between cold – dry periods and mild – wet periods appears to be governed by the North Atlantic Oscillation (NAO) and East Atlantic-West Russia (EAWR) teleconnection patterns (Oguz et al., 2006). The Black Sea exhibits a close coupling between anthropogenic and climatic forcing, as seen driving the dramatic ecosystem changes that were observed during the 1980s and 1990s (Oguz et al., 2006). The general circulation of the Black Sea is dominated by the persistent Rim Current, which circulates counterclockwise along the shelf break and horizontally mixes water masses throughout the whole basin (Oguz et al., 2005). Several coastal eddies are part of the Rim Current System and provide additional mixing across the shelf.

The north-western shelf is wider than elsewhere in the Black Sea and is substantially influenced by the discharge of several rivers (Dnipro, Dniester) of which the Danube is the most significant. These rivers transport sediments into the coastal zone, and particularly the Danube River built up a large Delta that spreads out into the Black Sea (Panin et al., 2016; Constantinescu et al., 2023). Additionally, the Danube is the largest source of freshwater to the Black Sea, and the discharge intensity directly affects the salinity gradient and hence stratification in the surface water in the western the Black Sea. The degree of stratification controls the vertical mixing, which controls the ventilation of the deep water with oxygen and also the replenishment of N and P in the euphotic surface zone. Altogether, these factors govern the susceptibility of Black Sea biogeochemical cycles to climate forcing: The warm and wet Atlantic climate increased the discharge of freshwater, thereby intensified stratification, and resulted in an upward shift of the oxycline. It reduced the availability of nutrients in the surface water of the central Black Sea, but increased availability of riverine nutrients in the river plume (Fulton et al., 2012). Conversely, low discharge of freshwater during the Sub-Boreal climate resulted in a deep oxycline and intensified upwelling of deep water that is rich in phosphorous and depleted in nitrogen (Fulton et al., 2012). Upwelling of low N and high P deep water to the surface reduces the molar $N:P$ ratio and favours diazotrophic $N_2$-fixation. Overall, this reduces the proportion of riverine N that fuels primary production (Fuchsman et al., 2008).

The Danube River is the second-largest river in Europe and drains a vast catchment area, which has been intensively used by humans since several millennia, making the Danube a significant source of nutrients to the north-western shelf. The increase in European population with spread of agricultural activities peaked first around 250 yr AD, and resulted in significant deforestation in central Europe, which caused erosion and hence the growth of river deltas (Maselli and Trincardi, 2013; Kaplan et al., 2009). This increased sediment transported towards the sea also led to increased nutrient transport and hence, pre-industrial eutrophication. The recent eutrophication of Danube accelerated in from 1960 to 1990 CE, during the "green revolution"), which resulted in a significant eutrophication of the north-western Black Sea Shelf (Kovacs and Zavadsky, 2021). The collapse of east-European economies after 1990 CE and later remediation measures led to a substantial decrease of the Danube load of dissolved inorganic nitrogen (DIN), which is now below the level of 1960 (Kovacs and Zavadsky, 2021). Möbius and Dähnke (2015) investigated present-day nutrient inputs to the shelf and argued that the majority of riverine DIN today is taken up by primary producers in the river plume and that the river nitrogen load is exported to the shelf as organic matter.

In this paper, we use nitrogen isotopes to identify nitrogen sources to the Black Sesa Shelf. Analysis of stable isotopes is a versatile tool as it provides distinct isotopic signatures (Kendall et al., 2007), which are expressed in the delta notation in the following. The delta notation describes the isotopic ratio of an element in a sample (e.g. $^{15}N/^{14}N$) in relation to the isotopic ratio in a standard material, and it was designed to conveniently express the variability of isotopic ratios in natural systems with small isotopic variation (McKinney et al., 1950). Nitrogen in ammonium and nitrate from fixation of atmospheric $N_2$ is isotopically light with $\delta^{15}N$ values around 0‰ (Zhang et al., 2014). This signature is preserved in phytoplankton as it assimilates dissolved DIN to produce organic nitrogen compounds. However, molecules with the lighter $^{14}N$ tend to diffuse and react slightly faster than molecules with the heavier $^{15}N$, which results in kinetic fractionation and gradually increases the relative concentration of $^{14}N$ in the product while $^{15}N$ is enriched in the remaining substrate. Consequently, the initial isotope signature evolves as the nitrogen is propagating though different pools. These fractionation effects accumulate in serial turnover, so that ammonium in soils is isotopically enriched with $\delta^{15}N$ values in the range of 5‰ to 10‰, while nitrogen in manure and sewage can reach nitrogen isotope values of up to 25‰ (Kendall et al., 2007). Since the isotopic signature of a nitrogen pool reflects the combined effects of its history (i.e., sources, turnover, and mixing), conclusions about all these parameters can be drawn from isotopic analyses. Johannsen et al. (2008) and Bratek et al. (2020) demonstrated that the $\delta^{15}N$ value of riverine nitrate is closely related to the intensity of anthropogenic land use. Similarly, Anderson and Cabana (2006) demonstrated that the $\delta^{15}N$ value of riverine nitrate is also related to the DIN load. Dähnke et al. (2008) and Serna et al. (2010) used the $^{15}N$ signature of anthropogenic nitrogen in sediments to identify the historical

onset of human-induced eutrophication in the North Sea and Skagerrak regions. In the following, we will investigate the isotopic enrichment of organic matter in sediment from the Danube-influenced Black Sea shelf to reconstruct nitrogen sources to this region. Specifically, we combine observations on N isotopes and nitrogen content to identify N sources and turnover processes. Briefly, if the $\delta^{15}$N value of organic matter changes but the N content remains stable, this can indicate a change in the N source, whereas an increase in $\delta^{15}$N and a concomitant decrease in N content is indicative of remineralisation (Möbius, 2013 TS4).

The present study aims to identify present and historic nitrogen sources to the Danube-influenced north-western (NW) shelf of the Black Sea by analysing sediment cores along a transect from the Danube Delta towards the shelf break. Similar studies by Fulton et al. (2012) and Cutmore et al. (2025) focussed on sediment from the deep basins and the continental slope of the Black Sea but did not cover the north-western shelf, where major rivers discharge into the Black Sea. Aiming to close this gap, we sampled along a transect from the Danube Delta towards the shelf break. Our samples reflect a gradient from Danube River Plume dominated to Black Sea-dominated water masses, which both imprint the specific signature of their respective nitrogen sources to the sediment record. We analysed the sediment for organic carbon and nitrogen, and the nitrogen stable isotope composition to identify natural and anthropogenic nitrogen sources over the past 6000 years.

## 2 Material and Methods

### 2.1 Working area and samples

Sampling was performed in early May 2016 during R/V *Mare Nigrum* cruise MN 148 in the Romanian Shelf area at four stations that span a transect from nearshore to offshore (Table 1, Fig. 1). Water depth at the sampling stations ranged from 22 m TS5 (Station 2) to 80 m (Station 6). From each station, sediment cores (20–40 cm length, 6 cm in diameter) were taken with a Multicorer. The sediment cores were immediately sliced in 1 cm intervals and frozen for further analysis. The sediment from stations 4 and 6 was wet sieved through a 400 µm sieve after slicing to collect mussel shells for radiocarbon dating. The $< 400$ µm fraction was freeze-dried and homogenized for analysis of $\delta^{15}$N, organic carbon and nitrogen content.

### 2.2 Analyses of sediment samples

The sediment samples were analysed for total carbon and total nitrogen content with an elemental analyser (Carlo Erba NA 1500) via gas chromatography, calibrated against acetanilide. The total organic carbon content (TOC) was analysed after a threefold removal of inorganic carbon using 1 mol L$^{-1}$ hydrochloric acid. Sediment carbonate content

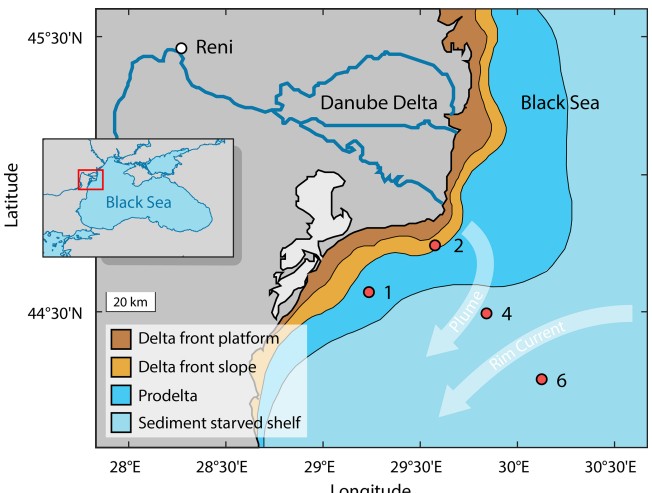

**Figure 1.** Sampling stations of the sediment cores 1, 2, 4 and 6 in the northwestern Black Sea during R/V *Mare Nigrum* cruise 148 and major depositional units of the Danube Delta (after Panin et al., 2016). Light arrows indicate the general surface water currents of Danube River Plume and Rim Current. Insert: Study area (red rectangle) within the Black Sea region.

**Table 1.** Summary of meta data of sediment cores from Mare Nigrum cruise 148.

| Core | Latitude TS6 | Longitude | Water depth (m) | Core length (cm) |
|------|----------|-----------|-------------|-------------|
| 1 | 45°58.4′ | 29°18.8′ | 30 | 42 |
| 2 | 44°74.9′ | 29°58.2′ | 22 | 35 |
| 4 | 44°49.9′ | 29°84.8′ | 62 | 29 |
| 6 | 44°25.2′ | 30°13.1′ | 80 | 27 |

was then calculated as the difference of total carbon content and TOC content. The standard deviation of sediment samples was less than 0.6 % for TOC and 0.08 % for nitrogen.

Nitrogen isotope analyses were performed with a CE 1108 elemental analyser (Thermofinnigan) connected to a mass spectrometer (Finnigan 252) via a split interface (Conflow). Two international standards were used for calibration (IAEA-N1: $\delta^{15}$N = 0.4 ‰, IAEA-N2: $\delta^{15}$N = 20.3 ‰), and an additional, internal standard was measured for further quality assurance. The standard deviation for repeated measurements was $< 0.2$ ‰.

### 2.3 Radiocarbon Dating

The radiocarbon ages of organic sediment (TOC) were obtained from 2 bulk sediment samples from Station 4, and 6 bulk sediment samples from Station 6. Additionally, 6 bivalve shells from different sediment layers of Station 6 (two samples of *Modiolula phaseolina* and four samples of *Mytilus galloprovincialis*) were analysed to date the carbonate. These two species were used because the top 8 cm

of the sediment are characterized by *Modiolula phaseolina*, whereas *Mytilus galloprovincialis* is dominant in deeper layers. The radiocarbon analyses were carried out at Beta Analytic Inc., UK, following standard procedures for accelerator mass spectrometry (AMS) radiocarbon dating. The radiocarbon ages are corrected for $\delta^{13}C$. Radiocarbon ages were calibrated to years before present (0 a $BP_{1950}$ = 1950 CE) using the Marine20 calibration curve (Heaton et al., 2020). The sample ages were further corrected with a reservoir age of $-111 \pm 63$ years ($N = 5$), based on data from Romanian and Bulgarian shelf sediment as provided by the Marine Reservoir Correction Database (Reimer and Reimer, 2001). The age of sediment samples between dated samples was estimated by linear interpolation.

## 2.4   $^{210}$Pb, $^{137}$Cs Dating

For $^{137}$Cs, $^{226}$Ra, and $^{210}$Pb measurement low-level gamma spectrometry was used. Sample preparation was carried out as described in Bunzel et al. (2020). Briefly, the cores were sectioned into slices of 1 cm thickness and frozen during transportation and storage. Each section was dried and homogenized by a ball mill. Aliquots of each sample were sealed in gas-tight Petri dishes and stored for minimum 28 d for equilibration of Radium-226 with daughter isotopes $^{222}$Rn, $^{214}$Pb and $^{214}$Bi. Measurements were performed by a high-purity low-level germanium detector (BE 3830P-7500SL-ULB Mirion Technologies/Canberra, Ruesselsheim, Germany). Measurement times varied between 90 000–600 000 s depending on sample activity. For calibration an artificial reference material was prepared from silica gel and reference solutions of $^{137}$Cs and $^{226}$Ra (Eckert & Ziegler Nuclitec GmbH, Braunschweig, Germany). Sediment ages were calculated from $^{210}$Pb results according to the CRS model (Appleby and Oldfield, 1978), assuming a constant rate of supply of atmospheric $^{210}$Pb. For consistent use of units, all $^{210}$Pb dating results are stated in years before present with 1950 CE as 0 (in a $BP_{1950}$).

## 2.5   Data integration and analyses

Sediment ages of cores 1 and 2 are based on results of Constantinescu et al. (2023), which sampled the same stations simultaneously and applied $^{210}$Pb and $^{137}$Cs dating. Observed Danube DIN loads are based on data from Kovacs and Zavadsky (2021), which presented DIN loads at Reni station at the upstream margin of the Danube Delta (Fig. 1). Both datasets were mapped to the corresponding sediment depths of cores 1 and 2 by linear interpolation. As the result, an interpolated $^{210}$Pb/$^{137}$Cs age and an interpolated DIN load was assigned to each of our sediment measurements of cores 1 and 2.

Using the interpolated DIN load and measured sediment N concentration, we derived two linear models from the DIN load – sediment N content correlation: Model 1 without $y$ intercept and Model 2 with $y$ intercept. From the DIN load – $\delta^{15}$N correlation, we derived Model 3.

The apparent isotopic fractionation factor ($\varepsilon$) was calculated by means of Rayleigh plots (Möbius, 2013). From the analysed subset of sediment samples, we used the largest measured total N content as reference for calculation of the remaining N fraction ($f$), which consequently plots at the coordinate origin.

Based on plots of $\delta^{15}$N vs. N content, we visually identified sediment layers with similar conditions. The underlying assumption was that in periods with a roughly constant trend in $\delta^{15}$N vs. N content, one distinct environmental condition dominated.

## 2.6   Data transformation

The age data in Fig. 7 were log transformed to emphasise the results from the most recent centuries. The data include age values after 1950 CE, which have a negative sign on the $BP_{1950}$ scale and cannot be log transformed. All plotted age data were thus converted to the $BP_{2020}$ scale where 0 a $BP_{2020}$ refers to the year 2020 CE. The axis labels correspond to the $BP_{1950}$ scale, so that when reading data from the diagram, the age data is displayed in the $BP_{1950}$ scale.

## 3   Results

### 3.1   Radioisotope measurements and dating

Sediment organic matter at Station 4 was dated $1035 \pm 97$ a BP at 7.5 cm core depth and $2837 \pm 98$ a BP at 16.5 cm core depth by radiocarbon ($^{14}$C) dating. The age of organic sediment at Station 6 spans from $137 \pm 93$ a BP at the sediment surface to $5679 \pm 104$ a BP at 16.5 cm depth. Radiocarbon-based ages of bivalve shell carbonates at Station 6 span from $3504 \pm 108$ a BP at the sediment surface to $5880 \pm 110$ a BP at 16.5 cm depth (Table 2, Fig. 2). At Station 6, dated carbonates were systematically older than the organic sediment, and this difference was larger at the sediment surface than at depths (Table 2, Fig. 2). Ultimately, organic sediment and carbonate shells represent two different carbon pools, which are affected individually by early diagenesis. In the following, we will focus on the organic sediment. No radiocarbon measurements were performed on sediment of Station 1 and 2.

Additional to $^{14}$C, we measured $^{137}$Cs and unsupported $^{210}$Pb in Station 4 sediment, where unsupported $^{210}$Pb was highest at the sediment surface (306 Bq kg$^{-1}$ dry sed. TS7) and decreased exponentially with depth. $^{210}$Pb was below detection limit below 4 cm sediment depth (Fig. 2). The estimated sediment ages ranged from $-75$ a BP at 0.5 cm sediment depth to 44 a BP at 3.5 cm sediment depth. Similarly, $^{137}$Cs activity was highest at the sediment subsurface (79 Bq kg$^{-1}$ dry sed.), but was detectable to deeper sediment

**Table 2.** Results of radiocarbon dating of organic sediment and carbonate shells from Cores 4 and 6, and calibrated age $\pm\,1\,\mathrm{SD}$, using Marine20 and $\Delta R = -111 \pm 63$ a. 0 a BP equals 1950 CE.

| Core | Sediment depth (cm) | Material | Conventional $^{14}C$ age (a) | Calibrated age (a BP) |
|---|---|---|---|---|
| 4 | 7.5 | organic sediment | $1530 \pm 30$ | $1035 \pm 97$ |
| 4 | 16.5 | organic sediment | $3080 \pm 30$ | $2837 \pm 98$ |
| 6 | 0.5 | organic sediment | $560 \pm 30$ | $137 \pm 93$ |
| 6 | 4.5 | organic sediment | $2180 \pm 30$ | $1727 \pm 109$ |
| 6 | 7.5 | organic sediment | $3550 \pm 30$ | $3408 \pm 108$ |
| 6 | 11.5 | organic sediment | $4280 \pm 30$ | $4348 \pm 121$ |
| 6 | 16.5 | organic sediment | $4790 \pm 30$ | $5004 \pm 126$ |
| 6 | 19.5 | organic sediment | $5380 \pm 30$ | $5679 \pm 104$ |
| 6 | 0.5 | carbonate (*Modiolula phaseolina*) | $3630 \pm 30$ | $3504 \pm 108$ |
| 6 | 7.5 | carbonate (*Modiolula phaseolina*) | $3490 \pm 30$ | $3334 \pm 105$ |
| 6 | 7.5 | carbonate (*Mytilus galloprovincialis*) | $4380 \pm 30$ | $4484 \pm 123$ |
| 6 | 11.5 | carbonate (*Mytilus galloprovincialis*) | $4650 \pm 30$ | $4823 \pm 121$ |
| 6 | 16.5 | carbonate (*Mytilus galloprovincialis*) | $5380 \pm 30$ | $5679 \pm 104$ |
| 6 | 19.5 | carbonate (*Mytilus galloprovincialis*) | $5570 \pm 30$ | $5880 \pm 110$ |

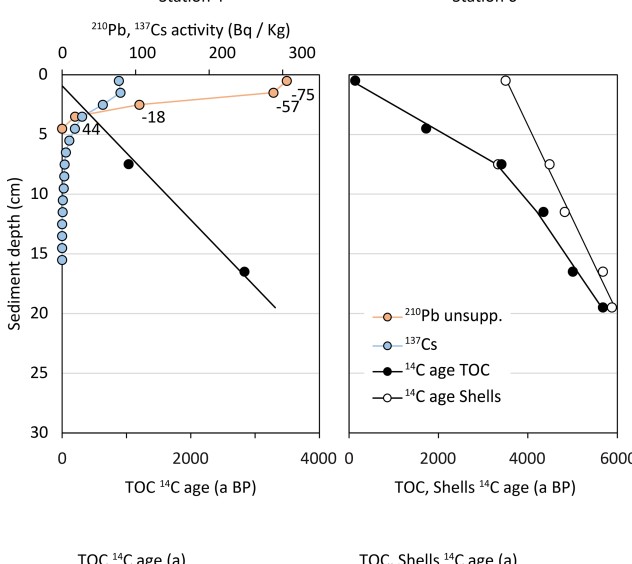

**Figure 2.** Results of radioisotope analyses of cores 4 and 6: Radiocarbon ($^{14}C$) ages of organic sediment (black circles), radiocarbon ages of carbonate shells (white circles). Measurements of unsupported $^{210}Pb$ (orange circles) and $^{137}Cs$ (blue circles). Numbers along $^{210}Pb$ plot indicate sediment age according to CRS model. No $^{210}Pb$ or $^{137}Cs$ data are available for core 6.

layers than unsupported $^{210}Pb$ (Fig. 2). No $^{210}Pb$ and $^{137}Cs$ measurements of sediment from Station 6 are available.

## 3.2 Sediment cores

The characteristics of sampled sediment reflect the proximity of the respective stations to the shore and the Danube Delta. Station 1 was in the shallow Prodelta (Fig. 1), where the sediment was layered mud with various shades of grey, and black layers at 28, 37, 41, and 44 cm sediment depth. Living *Mytilus* bivalves were found at the top layer and empty *Mytilus* shells within the black layers. Core 2 was sampled from the delta front slope (Fig. 1), and sediment was layered mud with various shades of beige, grey, and black. The sediment record of core 2 is affected by the sharp increase of sand content in some layers, which are the result of increased transport of sand from the Sfântu Gheorghe branch due to cutting off all meanders in Sfântu Gheorghe between 1984 and 1988 (Constantinescu et al., 2023). This led to an accelerated flow, riverbed erosion, and transport of coarser sediment in the main channel of the branch. Stations 4 and 6 are located on the distant shelf (Fig. 1) where sedimentation rates are lowest. In core 4, the top 0–3.5 cm were *Modiolula* shells in grey mud, followed by grey mud without shells down to 6 cm, and light grey mud from 6 to 26 cm sediment depth. Similarly, in core 6, *Modiolula* shells in mud were found at the top 0–5 cm, followed by light grey mud from 5 to 10 cm, and grey mud from 10 to 15 cm sediment depth. In 15 to 20 cm sediment depth, we found dark grey mud, and black mud with *Mytilus* shells in 20 to 25 cm depth.

## 3.3 Bulk sediment characteristics

Generally, the N isotope composition of sediment ($\delta^{15}N$) follows a gradient from the shore towards to open Black Sea: The entire nearshore sediment cores of stations 1 and 2 are isotopically enriched (mean $\delta^{15}N = 6.7\,\%_o$) with respect to atmospheric $N_2$ ($\delta^{15}N = 0.0\,\%_o$), while the distant stations were less enriched and were as light as $1.6\,\%_o$. The nearshore sediment cores had a lower concentration of organic matter than the more offshore cores.

Sediment from stations 1 and 2 was distinct from stations 4 and 6. While sediment close to the delta at stations 1 and 2 consisted of silt, silty clay and very fine sand, the sediment starved shelf station 4 and 6 sediment consisted of shelly clay (*Modiolula* and *Mytilus*). In detail, sediment at station 1 had low contents of TOC (1.2 % d.w.–2.9 % d.w.), TIC (1.1 % d.w.–2.1 % d.w.), and N (0.12 % d.w.–0.30 % d.w.). While TOC and N had a maximum in 17 cm sediment depth, TIC had no discernible variation with sediment depth. The molar TOC/N ratio decreased from 15 at the sediment surface to 8 at 18 cm sediment depth and increased again to 10 at 35 cm. Sediment at station 2 was similarly low in TOC, TIC, and N content had only small variation with sediment depth. TOC contents were in the range of 0.7 % to 1.9 % dry weight, the TIC contents in the range of 1.3 % to 1.8 % dry weight, and N contents in the range of 0.05 % to 0.26 % dry weight. The molar TOC/N ratio increased significantly from 8 at 35 cm sediment depth to 15 at the sediment surface, while no significant variation in $\delta^{15}$N values was observed (Fig. 3). The stations 4 and 6 on the continental slope were markedly different from stations 1 and 2. At station 6, small shell fragments of the bivalve *Modiolula phaseolina* were present in the upper 8 cm. Below 8 cm depth, bivalve shells of *Mytilus galloprovinialis* were found. At these stations, the organic carbon and nitrogen content decreased towards the surface in the upper 3 to 7 cm of the cores (TOC 1.1 % to 4.0 %; N 0.10 % to 0.42 %, Fig. 3). At station 4, we found a slight increase in organic carbon (1.0 % to 2.5 %) and nitrogen (0.10 % to 0.30 %) content below 7 cm depth and downwards. The TOC and TN contents strongly increased with depth at station 6 (TOC: 2.7 % to 9.2 %, N: 0.3 % to 1.0 %). In contrast, the molar TOC/N ratios decreased with sediment depth at station 4 and slightly increased with depth at station 6 (Fig. 3K, L).

### 3.4  N isotope signatures

Based on the $\delta^{15}$N vs. N content plots of Fig. 4, we identified 4 zones with distinct trends in N content and $\delta^{15}$N value. At the sediment surface, N content increased towards the sediment surface and N isotopes were most enriched (Fig. 4, filled circles), and we refer to this sediment layer as Zone 1 in the following. Zone 1 comprised the whole sampled sediment column at the coastal stations 1 and 2, and sediment in this layer had $\delta^{15}$N values in the range of 6 ‰–7 ‰ and an N content around 0.2 % (Fig. 4A, B). In the deeper stations 4 and 6, $\delta^{15}$N values were in the range of 4 ‰–6.5 ‰ and the N content was roughly around 0.3 % (Fig. 4C, D). Zone 1 reaches back until approx. 900 a BP (Fig. 4C, D). Below the sediment layer of Zone 1, the trend of $\delta^{15}$N vs. N content changed clearly. $\delta^{15}$N values still increased towards the surface, but the N content decreased, and we refer to this sediment layer as Zone 2. The $\delta^{15}$N values increased from around 2 ‰ to 4 towards the surface, while the N content decreased from ∼ 0.4 % to 0.2 % (open circles in Fig. 4C,

D). Since the trend of $\delta^{15}$N vs. N content in the sediment layer indicates kinetic fractionation by remineralisation, data from this layer were further analysed for the apparent isotope enrichment factor ($\varepsilon$) by means of Rayleigh plots. The estimated values were $\varepsilon = -1.1 \pm 0.2$ ‰ for Station 4, and $\varepsilon = 3.0 \pm 0.3$ ‰ for Station 6, respectively (Fig. 5). In core 4, this Zone went back to 4.9 ka BP, and in core 6 back to 4.3 ka BP.

The Zones 3 and 4 were only present at station 6. Sediment in the layer of Zone 3 was characterized by a constant N content of approx. 0.6 % while $\delta^{15}$N values were decreasing from ∼ 3 ‰ down to 1.6 ‰ (Fig. 4D, closed triangles). This layer was dated 5.0 to 4.3 ka BP. Zone 4 is characterised by N contents that decreased from approx. 1.1 % down to 0.6 %, while $\delta^{15}$N values were constant around 3.3 ‰ (Fig. 4, open triangles). Zone 4 comprised sediment from the bottom end of the core with an age of 6.9 ka BP to sediment with an age of 5.0 ka BP.

### 3.5  Danube DIN load models

The correlation of Danube DIN loads with sediment N content and $\delta^{15}$N in the nearshore cores 1 and 2 at a given time was examined to develop a simple empirical model to reconstruct historical DIN loads for the period before measurements were available. Using core 2, no meaningful correlation was found (not shown). Using core 1, the DIN load of Danube at Reni station (Kovacs and Zavadsky, 2021) correlated significantly with the bulk N content (Model 1: $R^2 = 0.98$, Model 2: $R^2 = 0.64$) and less significant with $\delta^{15}$N ($R^2 = 0.24$, Model 3). The N content-based Models 1 and 2 had average residuals with respect to observed DIN loads of $36 \pm 26$ and $42 \pm 31$ kt yr$^{-1}$, respectively (Fig. 6B, D). The average residuals of the $\delta^{15}$N-based Model 3 were $61 \pm 33$ kt yr$^{-1}$ (Fig. 6F). For the period 1800–1950, all three models estimated that the Danube DIN load was 236 to 318 kt yr$^{-1}$ in 1800 CE and increased gradually with 0.2 to 0.5 kt yr$^{-1}$.

## 4  Discussion

### 4.1  Overview

Based on $\delta^{15}$N and N content measurements of sediment from the NW shelf we identified four distinct sediment layers where each one was characterised by a distinct combination of $\delta^{15}$N and N content dynamics (Fig. 4). We thus assume that these four layers represent the record of distinct conditions on the NW Black Sea shelf. In the following, we will interpret the data from these four Zones and discuss the implications for the major nitrogen sources that drive primary productivity on the shelf during the respective periods.

We start by combining data from cores 1, 4, and 6 into a joint plot to construct a composite timeline plot (Fig. 7) and interpret the sediment record imprinted in the northwest-

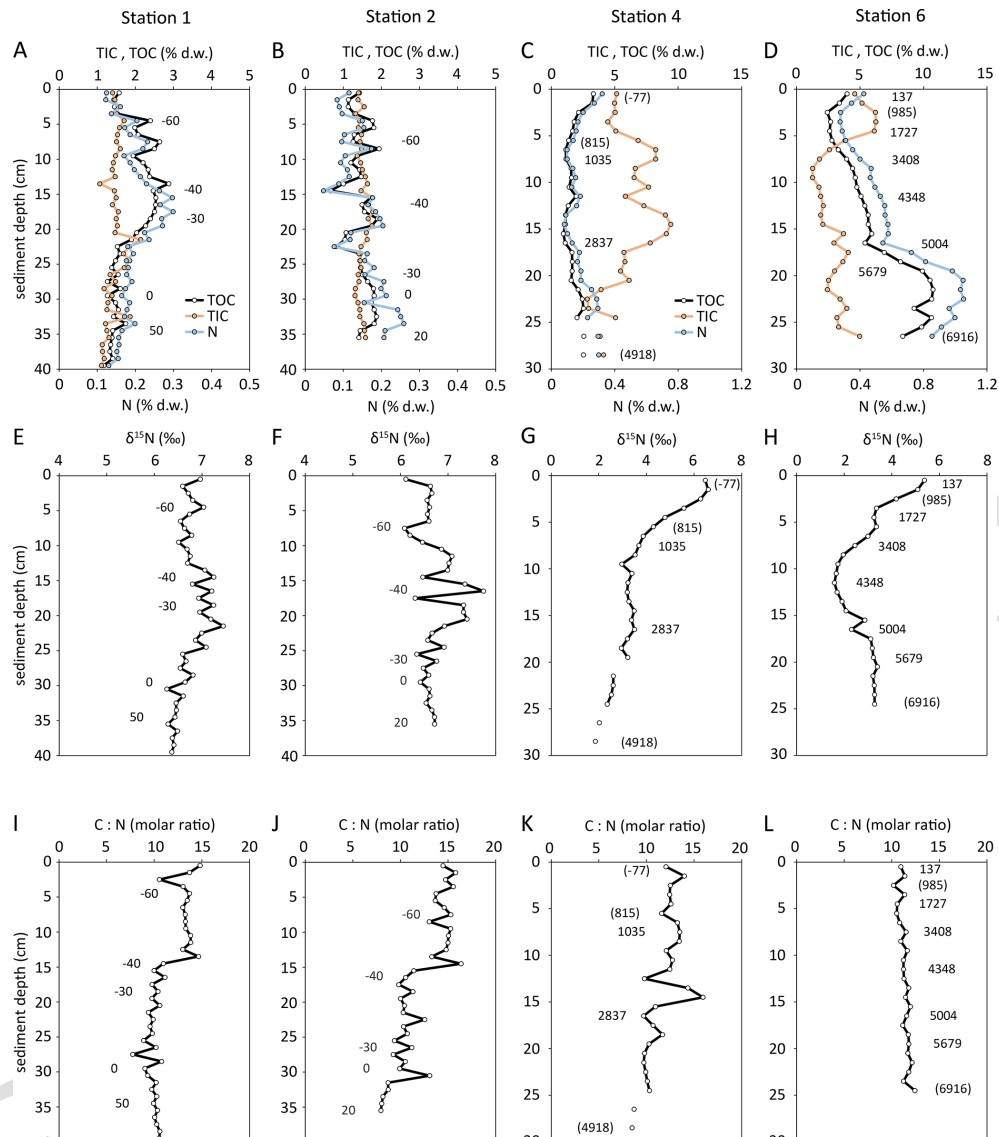

**Figure 3.** From Stations 1–6, depth profiles of TIC (carbonate), TOC (organic sediment), total nitrogen **(A–D)**, measured $\delta^{15}N$ values of bulk sediment **(E–H)**, and molar TOC/N ratio of organic sediment **(I–L)**. Numbers refer to the sediment age BP (0 a BP = 1950 CE, negative age values are after 1950 CE, positive age values are before 1950 CE), based on $^{210}$Pb (Stations 1, 2) and $^{14}$C (Stations 4, 6). Numbers in parentheses were estimated by linear interpolation. $^{210}$Pb data from Constantinescu et al. (2023).

ern shelf. We excluded data from core 2 due to the absence of correlation of Danube DIN load and sedimentary N content and $\delta^{15}N$ (see results, Sect. 3.3 TS8 ). The individual $\delta^{15}N$ plots match well where the plots overlap, and the continuity of the composed $\delta^{15}N$ plot suggests that organic matter in the water column was mixed across the entire shelf prior to deposition on the sediment (Fig. 7). Similarly, we combined the N content data from cores 1, 4, and 6, and found systematic offsets between the cores (Fig. 7) in the sense that sediment farther from the delta had higher N content than sediment from the same period that was deposited closer to the delta. However, we still found simultaneous variations of N content

over time, which we interpret as a result of higher deposition rates of terrigenous material closer to the delta. The corresponding sedimentation rates of terrigenous matter were up to 10 mm yr$^{-1}$ close to the delta (Constantinescu et al., 2023) and as low as 0.03 mm yr$^{-1}$ (Table 2) on the deeper shelf. The higher sedimentation rates near the delta effectively diluted the deposited organic matter more than the low sedimentation rates did farther from the delta.

To elucidate how variations in climate forcing, stratification of the water column, and human activity are reflected in the sediment record of the NW shelf, we now turn to our

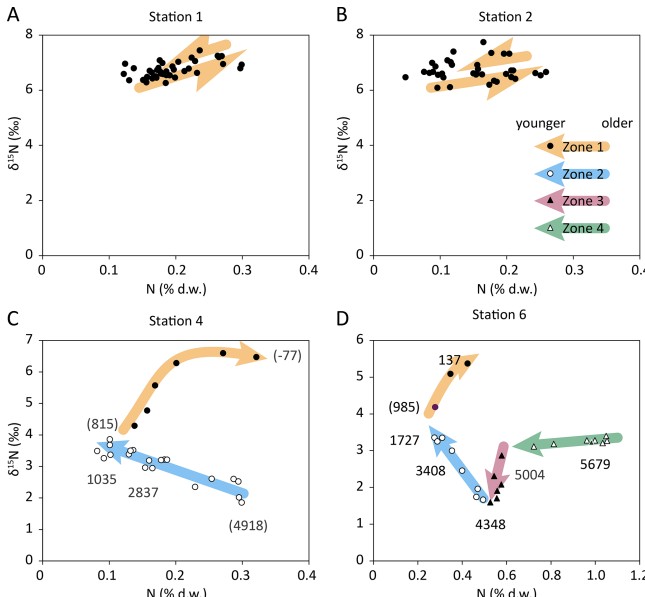

**Figure 4.** Sedimentary $\delta^{15}$N vs. sedimentary nitrogen content at Station 1 (**A**), Station 2 (**B**), Station 4 (**C**), and Station 6 (**D**). The different symbols indicate different process zones within the sediment column: (Zone 1) filled circles indicate modern eutrophication, (Zone 2) open circles indicate diagenetic enrichment, (Zone 3) filled triangles indicate the gradual transition between two isotopically distinct nitrogen sources, and (Zone 4) open triangles indicate Unit II sapropel. Numbers refer to the $^{14}$C-based sediment age, numbers in parentheses were estimated by linear interpolation. Coloured arrows represent the arrow of time for distinct trends of $\delta^{15}$N vs. N content.

observations from the four distinct $\delta^{15}$N/N Zones (Fig. 4) in detail.

### 4.2 Strong stratification and sapropel formation: 6.9–5.7 ka BP

The oldest sediment in our cores dates back 6.9 ka BP at station 6. At that time, the Black Sea was influenced by the humid Atlantic climate with high freshwater input from rivers and seawater influx from the Mediterranean through the Bosphorus, leading to a strong thermohaline stratification and a shallow chemocline. The shallow chemocline at that time was confirmed by Cutmore et al. (2025) through the presence of isorenieratene in the sediment, which is an indicator of a shallow chemocline that allows hydrogen sulphide to ascend into the photic zone. These euxinic conditions can substantially constrain the degradation of sinking particles in the water column and in the sediment once they have been deposited. As a result, measured isotopic values of sediment from this period should be close to the original isotopic signature. We indeed measured a $\delta^{15}$N value of 3.3 ‰ in this sediment layer, which is consistent with riverine N from a pristine catchment with no anthropogenic land use (Johannsen et al., 2008; Bratek et al., 2020) and thus in-

dicates that riverine nitrogen was the dominant N source to the NW shelf. For the same period, Fulton et al. (2012) reconstructed a $\delta^{15}$N range of phytoplankton from 1.4 ‰ to 4.4 ‰, and our results agree with this reconstruction. We thus conclude that the N/$\delta^{15}$N trend of Zone 4 (Fig. 4 open triangles, Fig. 7) indicates a sapropel with high organic matter concentration that was deposited in the Black Sea basin during this phase and was termed stratigraphic unit II b (Ross et al., 1970). Our data from this period show similarly high TOC values in core 6 and a matching $\delta^{15}$N value of 3.3 ‰ (Fig. 3).

### 4.3 Shift to Nitrogen Fixation: 5.0–4.4 ka BP

The humid Atlantic phase lasted until approx. 5.1 ka BP and was superseded by the Sub-Boreal climate, which was colder and dryer. This period resulted in substantially reduced riverine input and a weakened salinity gradient due to increased surface salinity (Giosan et al., 2012). The weaker stratification led to a deeper circulation, which enhanced pelagic ventilation (Fulton et al., 2012). We observed a clear change in bulk $\delta^{15}$N values and N content around 5.0 ka BP at the most offshore station 6 and found a similarly low sediment $\delta^{15}$N value of 1.9 ‰ at station 4 (Fig. 3, open circles). In this dry period, deep circulation may have favoured $N_2$-fixation (Fig. 7, Zone 3). We suggest that the N source gradually shifted from riverine N to $N_2$-fixation in the 5.0–4.4 ka BP period and is represented by the N/$\delta^{15}$N trend of Zone 3 (Fig. 4 closed triangles, Fig. 7). The sediment of this period is equivalent with sediment Unit II a, which is the upper part of the organic-rich sapropel layer (Ross et al., 1970). The intensified and deeper mixing of the water column did not only mix oxygen downwards but also enabled the upward transport of nitrogen depleted and phosphate enriched deep water into the surface water, which resulted in N : P ratios in the euphotic zone in the range of 3.5–6. This surplus of P thereby favoured $N_2$- fixation by cyanobacteria (Fulton et al., 2012). The reconstructed $\delta^{15}$N value of phytoplankton during this period was in the range of 0.3 ‰ to 2.1 ‰ (Fulton et al., 2012), and in combination with additional proxies this confirms the dominant role of $N_2$-fixation (Cutmore et al., 2025). $N_2$-fixation introduces the low isotope signature of atmospheric $N_2$ and results in plankton with a comparatively low $\delta^{15}$N value of approximately 1 ‰ to 2 ‰ (Minagawa and Wada, 1986). However, the $\delta^{15}$N values we observed in sediment from the NW shelf are slightly higher than those reported by Fulton et al. (2012) and Cutmore et al. (2025). The latter were sampled at greater water depth at locations farther from the Danube Delta. This difference between our data and previously published values suggests that isotopically heavy N from riverine inputs had a higher contribution to the N supply at the NW shelf than at more distant deeper regions of the Black Sea.

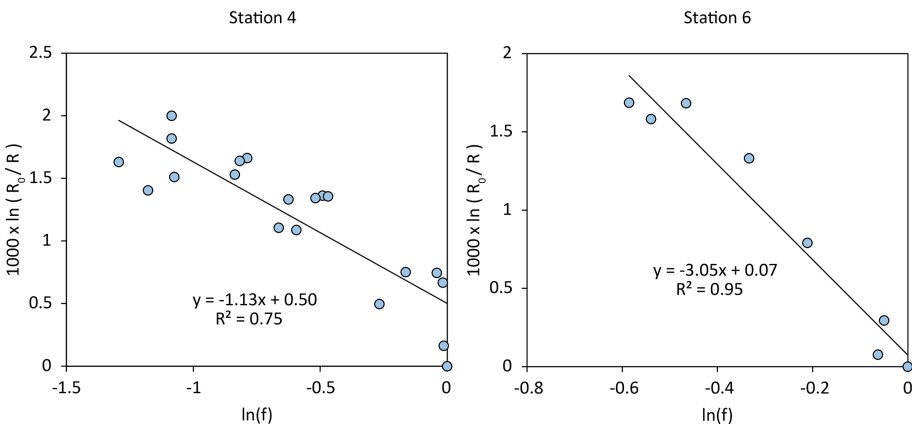

**Figure 5.** Rayleigh plots of $\delta^{15}$N vs. sedimentary N content of samples from Zone 2 of Stations 4 and 6 (see also Fig. 4C, D, open circles).

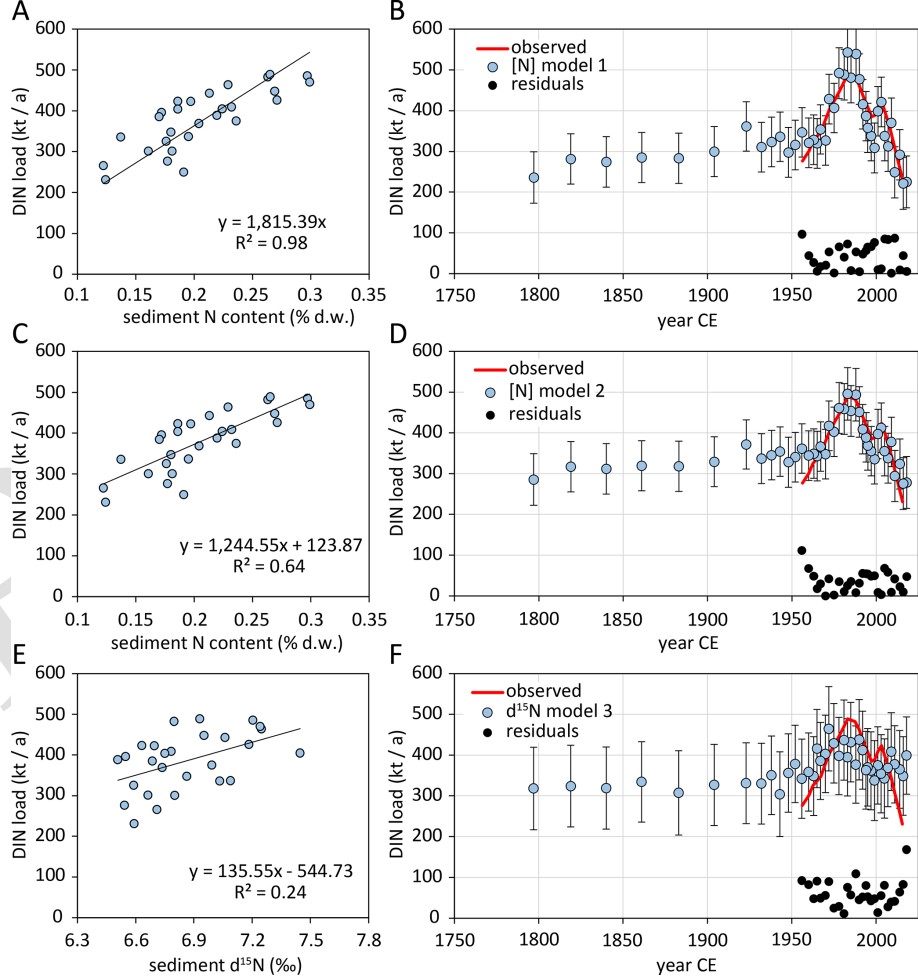

**Figure 6.** Linear correlations of Danube DIN loads with sediment N content at station 1 at a given time, based on sediment dating (Constantinescu et al., 2023) **(A, C)** and sediment $\delta^{15}$N values **(E)**, and reconstructed DIN loads based on these correlations **(B, D, F)**. Error bars indicate prediction intervals with 90 % confidence. Red lines indicate DIN observation data for 1955–2015, data from Kovacs and Zavadsky (2021).

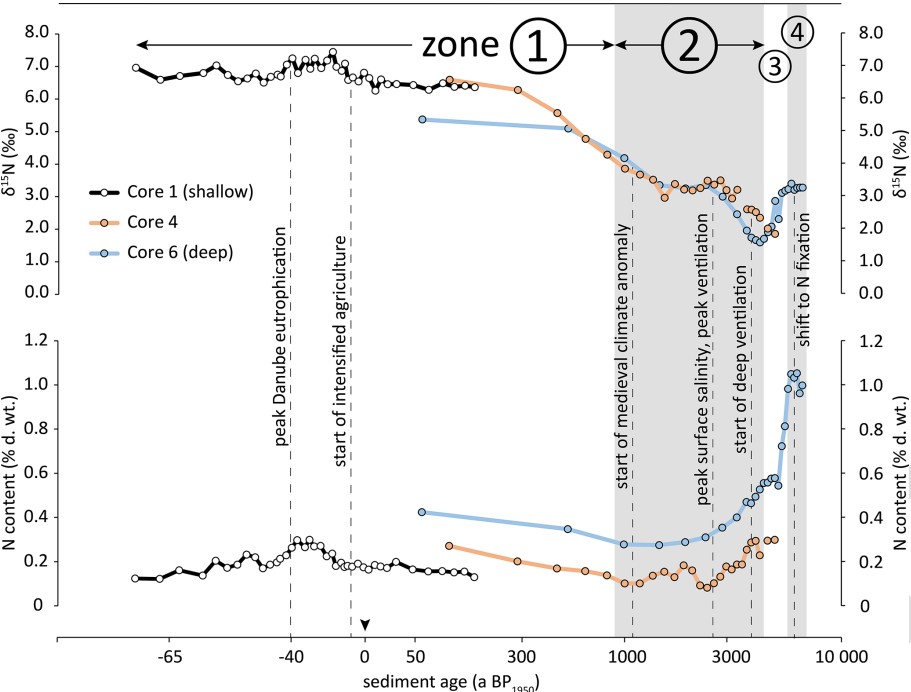

**Figure 7.** Evolution of N content and $\delta^{15}$N values in sediment cores 1, 4 and 6 from the NW shelf. Grey and white backgrounds indicate $\delta^{15}$N/[N] zones 1–4, additionally marked by circled numbers. Vertical dashed lines mark significant events mentioned in the discussion. Age scale is log transformed (see methods Sect. 2.6 TS9 for details). TS10

## 4.4 Oxygenated sediment at the shelf break: 4.4–0.9 ka BP

The intensified and deeper circulation during the Sub-Boreal phase resulted in an intensified ventilation of the shelf water as confirmed by Cutmore et al. (2025), who did not detect the proxy isorenieratene for $H_2$S in the photic zone during this period (3.9–2.7 ka BP in their study). Before and after this period, this proxy was always present and indicated an exceptional deep ventilation. This deeper ventilation of the water column gradually increased the exposure of shelf sediments to oxygen and thereby enabled enhanced remineralisation of deposited organic matter. Möbius et al. (2010) demonstrated that early diagenesis is indicated by increasing $\delta^{15}$N values and decreasing N concentrations, which we indeed found in cores 4 and 6 in the period 4.4 to 0.9 ka BP (marked as Zone 2 in in Figs. 4, 7). During this period, the highest $\delta^{15}$N value and lowest N concentration coincided with the peak of surface water salinity (Giosan et al., 2012). We interpret this as an indication of pronounced benthic remineralisation when the salinity gradient was weakest and thus ventilation was most intense (Fig. 7).

In core 4, the apparent enrichment factor for sedimentary nitrogen of $\varepsilon = -1.1 \pm 0.2\,‰$ falls well within the range of published values for remineralisation of organic matter (Möbius et al., 2010), and we thus assume that the observed increase in $\delta^{15}$N values in Zone 2 is rather a result of reminer-

alisation and not an indication of changes in nitrogen sources. However, we found a different situation in core 6, where the apparent enrichment factor for the same period was much higher ($\varepsilon = -3.0 \pm 0.3\,‰$), which is unusually high for remineralisation alone. We therefore interpret the isotopic variation in core 6 as the combined effect of early diagenesis and a gradual shift from $N_2$-fixation to isotopically more enriched riverine N input.

In summary, we conclude that between 4.4–0.9 ka BP station 4 was supplied by a quite stable mixture of N from river discharge and from pelagic nitrogen fixation. The more offshore station 6 was initially received isotopically depleted nitrogen from $N_2$-fixation (approx. 4.4 ka BP), which was gradually complemented by isotopically enriched river borne nitrogen until 0.9 ka BP (Fig. 4). This implies that the influence of the Danube River plume extended from station 4 to station 6 in this period. At around 1.0 ka BP, $\delta^{15}$N values of sediment from cores 4 and 6 were around 4 ‰, which is substantially above the values reported by Fulton et al. (2012) and Cutmore et al. (2025) for this period. They reported $\delta^{15}$N values of 1 ‰ and 0.5 ‰, respectively, from more distant locations. This difference underscores that the sediment record from the NW shelf reflects other processes and N sources than sediment from more distant parts of the Black Sea.

The occurrence of coccoliths from the haptophyte plankton algae *Gephyrocapsa huxleyi* (formerly *Emiliania huxleyi*) in the sediment starts approximately at 3.6 ka BP (Hay et al.,

1991; Coolen, 2011) and thus also falls into the period 4.4–0.9 ka BP. We found a corresponding increase in TIC in sediment from cores 4 and 6 from 3.6 ka BP onwards until approx. 0.9 ka BP (Fig. 3). Since low N : P ratios in water do not only favour $N_2$-fixation but also promote blooms of *G. huxleyi* (Lessard et al., 2005), the presence of coccoliths can indicate that the outer shelf was influenced by an N-deficit and thus by $N_2$-fixation.

### 4.5 Anthropogenic eutrophication and recovery: 900 a BP to present

At around $900\,\mathrm{a\,BP}_{1950}$, we observe an increase in $\delta^{15}N$ values and N content, which indicates that the condition changed on the NW shelf. The $\delta^{15}N$ values eventually exceeded the values from Zone 1, in which N from pristine rivers was the dominant N source. Instead, the high $\delta^{15}N$ values indicate the deposition of N that was isotopically enriched by human activities (Johannsen et al., 2008; Bratek et al., 2020). This deposition of substantially enriched N started around $900 \pm 120\,\mathrm{a\,BP}_{1950}$ in cores 4 and 6 (Fig. 7), and thus much earlier than the industrialisation in the 20th century with widespread use of artificial fertiliser. Fulton et al. (2012) and Cutmore et al. (2025) consistently found a significant increase of sediment $\delta^{15}N$ values in cores from the Black Sea shelf and deep basins. This increase started around $0.5\,\mathrm{ka\,BP}_{1950}$, which supports our observation that the deposition of enriched nitrogen began much earlier than the industrialisation. The difference of approximately 400 years between the onset of enriched nitrogen deposition on the Danube influenced shelf (this study) and the deeper locations further south (Fulton et al., 2012; Cutmore et al., 2025) most likely reflects differences in the sensitivity of these locations to signals from the Danube.

The early onset of isotopically enriched nitrogen deposition could be an artifact of bioturbation in which benthic macrofauna mixes modern, isotopically enriched nitrogen from the sediment surface downwards and thus into older sediment layers. However, the sediment cores 4 and 6 were populated by sessile tunicates and small bivalves (*Modiolula phaseolina*), which are no strong bioturbators and thus are unlikely to provide sufficient sediment mixing to transport anthropogenic $^{15}N$ down to 7 cm sediment depth (station 4). Our measurements of particle-associated $^{210}Pb$ further indicate that the mixed surface layer was no deeper than 4 cm at maximum (Fig. 2), which was significantly above the deepest occurrence of enriched nitrogen (7 cm depth, Fig. 3). The deeper penetration of $^{137}Cs$ does not contradict our interpretation, because $^{137}Cs$ has a higher mobility in marine sediment than $^{210}Pb$ and usually migrates into deeper sediment layers (Wang et al., 2022). Additionally, the carbonate content of cores 4 and 6 decreased simultaneously with increased $\delta^{15}N$, which is not a plausible result of sediment mixing by bioturbation. Instead, the decreasing carbonate content in the modern surface layer indicated a change in the nutri-

ent regime with a shift from coccolithophorid blooms to dinoflagellate blooms in the coastal area (Giosan et al., 2012).

One explanation for the early deposition of enriched nitrogen is intensified N discharge during the Medieval Warm Period/Medieval Climate Anomaly (Mann et al., 2009). During this local climate optimum in Europe between 1000 and $700\,\mathrm{a\,BP}_{1950}$, a substantial population growth led to an expansion of agricultural land use and urbanization and to substantial deforestation in Europe (Giosan et al., 2012). The Medieval Climate Anomaly indeed coincides with the onset of anthropogenic N deposition on the NW shelf (Fig. 7). A pre-industrial eutrophication of the Danube River is further supported by our reconstruction of Danube DIN loads for the 19th century (Fig. 6). The modelled DIN loads based on correlations of observed DIN load and sedimentary bulk nitrogen content suggest that the Danube DIN load was in the range of 236 to $286\,\mathrm{kt\,a^{-1}}$ in 1800 CE, comparable to the current DIN load (Kovacs and Zavadsky, 2021). Although the river DIN load was substantially less correlated with shelf sediment $\delta^{15}N$ values than with shelf sediment bulk N content, the reconstructed DIN load based on $\delta^{15}N$ yields similar results. Additionally, the average slope of the trend of the Danube River DIN load in 1800–1950 ($0.35 \pm 0.16\,\mathrm{kt\,yr^{-2}}$) can be linearly extrapolated approximately 800 years back until the modelled Danube River DIN load approaches zero. Although this extrapolation reaches very far into the past with respect to the relatively short period of underlying observational data and thus is tied to substantial uncertainty, our model results further support an early onset of anthropogenic eutrophication of the Danube. Our approach to reconstruct historical Danube River DIN loads relies on the assumption that quantity and isotopic composition of Danube River DIN translate linearly to Black Sea sedimentary N content or bulk $\delta^{15}N$ values, although the dissolved N is assimilated into phytoplankton, transported, deposited in the delta, and partially degraded by early diagenesis. The approach appears valid for the 1955–2015 period, and we are not aware of conflicting results to challenge our approximations of historic Danube DIN loads. The results of the offshore stations 4 and 6, which go far back into the past, in combination with results from the coastal station 1, which recorded the N deposition of the last 200 years in more detail, yield a coherent picture of the steadily increasing eutrophication of the Danube for at least $900 \pm 120$ years, which has only decreased in the last 30 years. The sediment record of Station 2 does not reflect these processes in sufficient detail likely due to its location on the active delta front slope (Fig. 1). The sediment there is affected by sand deposits from the Sfântu Gheorghe Danube branch as a results of cutting off all the meanders of Sfântu Gheorghe, between 1984 and 1988, which led to an accelerated flow in the main channel and scouring of its river bed (Constantinescu et al., 2023).

The youngest major event reflected in the sediment record is the massive eutrophication during the "Green Revolution" since the 1960s, with intensified discharge of nutrients, en-

hanced primary production and oxygen consumption due to enhanced organic matter decomposition in deeper water layers, and thus a shallower oxycline. The increased deposition of organic matter with isotopically enriched nitrogen is evident in all cores and is especially obvious in core 1. There, the highest N content and highest $\delta^{15}$N values coincide with the peak of eutrophication during the 1980–1990 period ($-30$ to $-40\,a\,BP_{1950}$, see Figs. 3, 7) and decrease simultaneously, in parallel to the dropping nitrogen load of Danube after the 1990s (Kovacs and Zavadsky 2021).

If our hypothesis holds true that the eutrophication of the Danube started several centuries before the onset of industrialisation, this further implies that Danube was not pristine in the sense of the European Water Frame Directive (WFD) since the Middle Ages. The WFD requests management of water bodies at the river basin level to achieve "good status" for all water bodies, which basically requests a condition with no or minimal human impact (European Commission, 2000), including the riverine nutrient load. The outcome of our study indicates that defining such "good status" of a water body may not be possible for some systems as environmental conditions are difficult to reconstruct. In contrast, it may be recommended to base good environmental status on the ecosystem functions of the water body and the ecosystems associated to it.

## 5 Conclusions

We sampled sediment across the NW shelf of the Black Sea across a gradient ranging from high influence of Danube River nutrient input to low influence of river input close to the shelf break. We analysed the nitrogen stable isotope composition and the nitrogen content of the sediment to identify nitrogen sources to the primary production on the NW shelf. Our results indicate that the relative contribution of riverine nitrogen and pelagic $N_2$-fixation fluctuated during the past 6000 years and that this fluctuation was largely driven by climate changes. Due to the proximity of our sampling sites to the Danube Delta, the sediment record was susceptible to signals from the Danube. The sediment record suggests that the deposition of isotopically enriched nitrogen, likely from human activities, started approximately 900 years ago. This deposition nitrogen from anthropogenic activities thus started surprisingly long before the onset of industrialisation, which is commonly believed to have induced the current eutrophication in the 20th century. Instead, the Danube was not pristine with respect to nutrient loads since the Middle Ages. Our reconstructed DIN loads suggest that already around 1800 CE, the Danube River eutrophication was at a similar level as today, and that DIN loads gradually increased throughout the 19th and 20th century until 1960 CE. Then, eutrophication steeply increased even further and peaked around 1990 CE due to intensified agriculture, the so-called Green Revolution. After 1990, the Danube

River DIN loads decreased significantly due to economic collapse in the early 1990s and nutrient reduction policies afterwards in the Danube River catchment, and this reduction of the Danube N-load is already reflected in the western Black Sea coastal sediment record.

*Code availability.* . TS11

*Data availability.* . TS12

*Author contributions.* Conceptualization: AN, AB, JEEvB, JF, JM, TS, KD; Formal analysis: AN, AB, JM, HW; Investigation: JEEvB, AB, JM, HW.
    Visualization: AN, Writing (original draft preparation): AN, Writing (review and editing): AN, JEEvB, JF, JM, TS, HW, KD.

*Competing interests.* The contact author has declared that none of the authors has any competing interests.

ther geographical representation in this paper. The authors bear the ultimate responsibility for providing appropriate place names. Views expressed in the text are those of the authors and do not necessarily reflect the views of the publisher.

*Acknowledgements.* We wish to thank the captain and the crew of the R/V *Mare Nigrum*. We are grateful to M. Ankele, M. Metzke and N. Lahajnar for analytical work. We further thank L. Hoffman for the taxonomic identification of bivalve shells. The IAEA is grateful to the Government of the Principality of Monaco for the support provided to its Marine Environment Laboratories. We appreciate the comments of three anonymous reviewers to further improve our manuscript. No so-called AI tools have been used for this study.

*Financial support.* This study was supported by the project ReCoReD (Reconstructing the Changing Impact of the Danube on the Black Sea and Coastal Region) funded by TNA FP7 EuroFleets 2, and by the DOORS project (European Commission, Horizon 2020 Framework Programme TS13, grant no. 101000518).

The article processing charges for this open-access publication were covered by the Helmholtz-Zentrum Hereon.

*Review statement.* This paper was edited by Perran Cook and reviewed by three anonymous referees.

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

**Remarks from the typesetter**

**TS11** Please provide a statement on how your underlying software code can be accessed. If the code is not publicly accessible, a detailed explanation of why this is the case is required. The best way to provide access to software code is by depositing it (as well as related metadata) in reliable public repositories, assigning digital object identifiers (DOIs), and properly citing code as an individual contribution. Please indicate if different software codes are deposited in different repositories or if code from a third party was used. Additionally, please provide a reference list entry including creators, title, and date of last access. If no DOI is available, assets can be linked through persistent URLs to the software code itself (not to the repositories' home page). This is not seen as best practice and the persistence of the URL must be secured.

**TS12** Please provide a statement on how your underlying research data can be accessed. If the data are not publicly accessible, a detailed explanation of why this is the case is required. The best way to provide access to data is by depositing them (as well as related metadata) in reliable public data repositories, assigning digital object identifiers (DOIs), and properly citing data sets as individual contributions. Please indicate if different data sets are deposited in different repositories or if data from a third party were used. Additionally, please provide a reference list entry including creators, title, and date of last access. If no DOI is available, assets can be linked through persistent URLs to the data set itself (not to the repositories' home page). This is not seen as best practice and the persistence of the URL must be secured.