# Peer review of "Reconstructing changes in nitrogen input to the Danube-influenced Black Sea Shelf during the Holocene"

_EGUsphere, 2025_

## Author Response (AR1)

egusphere-2025-1803
Title: Reconstructing changes in nitrogen input to the Danube-influenced Black Sea Shelf during the Holocene

**1) General remarks**

We appreciate the thorough review of our manuscript and the detailed advice from the reviewers. We have edited our manuscript accordingly, and we are confident that the manuscript is now significantly improved. Our detailed response is listed below.

**2) Reviewer comments and our responses.**

Review 1:

| Reviewer's comment | Our response |
|---|---|
| Containing large anoxic water body, Black Sea is likely affected by water column denitrification (WCD) and other N loss processes. WCDs can critically affected N isotope signals and nutrient pool and is likely to affect N sources analysis. The authors should include Black Sea WCD in the Introduction and Discussion section. | We added water column denitrification as a significant process of nitrogen turnover to the discussion (section 4.6): The relevant snippet is: "The turnover of nitrogen during remineralisation tends to alter its isotopic ratio, which raises the question to which degree the nitrogen in the organic matter deposited to the sediment corresponds to the nitrogen that was initially assimilated by phytoplankton. Degradation of plankton biomass in the water column initially releases isotopically light nitrogen as ammonium (Möbius & Dähnke 2015), which can subsequentially be oxidised to nitrate. Nitrate is prone to denitrification in anoxic conditions such as the sediment or the deeper part of the Black Sea, which would leave behind the nitrogen in the remaining organic matter with altered isotopic ratio. However, $H_2S$ in euxenic environments reduces the degradability of organic matter (Raven et al. 2018, Kok et al. 2000), which would also protect the isotopic signature of nitrogen therein. Additionally, Möbius & Dähnke (2015) found that the plankton community of the Danube River Plume efficiently assimilates nitrogen from the water |

| | and thereby outcompetes ammonium oxidising and denitrifying bacteria. This means that the plankton community efficiently keeps the nitrogen in particulate organic matter until it is eventually deposited on the sediment close to the Danube Delta and that water column denitrification is not a significant sink of nitrogen on the Danube-influenced shelf." |
| :--- | :--- |
| | We additionally added N-sinks to the Discussion. The relevant snippet is" The oxycline between the ventilated surface water and the euxenic deep water promotes substantial rates of N-loss by water column denitrification (Fuchsman et al. 2019) or anammox (Kuypers et al. 2007)." |
| Another concern is the novelty. Fulton et al., 2012 did some similar analysis regarding N fixation analysis in Black Sea with similar proxies. Authors should clarify what is the major novelty compared with Fulton et al., 2012 and prior research. | It is correct that Fulton et al. 2012 did a similar analysis. In our study we use Fulton et al. 2012 as a reference for our own results to ensure that our own results are consistent with previous observations. A novelty of our study is that we sampled the upper shelf at the Danube Delta while Fulton et al. have sampled the deep basins. While the sediment record of the deep basins integrates signals over the whole Black Sea, this is not necessarily the case in our study area where local processes can be dominant over basin wide signals. The most significant novelty is the reconstruction of Danube N loads with a high temporal resolution for the past 200 years, which was not attempted by Fulton et al. 2012. |
| | We now emphasize the differences between our results and results of Fulton et al. 2012 clearer in the discussion section. |
| | Relevant snippets from section 4.3: "However, the δ15N values we have observed in sediment from the NW shelf are slightly higher than the values observed by Fulton et al. (2012) and Cutmore et al. (2024), which have sampled locations farther from the Danube Delta and with deeper bottom depth. This offset hints to a higher contribution of isotopically heavier N |

| | from riverine inputs at the NW shelf." |
| | |
| | And from section 4.4: ". At around 1.0 ka BP, δ15N values of sediment from Cores 4 and 5 were around 4 ‰, which is substantially above the values of 1 ‰ reported by Fulton et al (2012) and 0.5 ‰ reported by Cutmore et al. (2024) and underlines that the sediment record from the NW shelf reflects different processes than the samples from deeper stations." |
| Line 73-75 Define delta N isotope notation first. | We added to the introduction: "Analysis of stable isotopes is a versatile tool as it provides distinct isotopic signatures (Kendall et al. 2007), which are expressed using the delta notation in the following. The delta notation expresses the isotopic ratio of an element in a sample (e.g. 15N/14N) in relation to the isotopic ratio in a standard material, and it was designed to conveniently express the variability of isotopic ratios in natural systems in which the range is very small (McKinney et al. 1950)." |
| Section 2.4 and 2.5: The age calculation methods based on Pb and Cs isotope should also be included in methods sections. | We added the Pb-210 dating method as the methods section 2.4: 210Pb, 137Cs Dating |
| Section 3.1: I did not see the age results from the Pb/Cs isotopes. I think the dating results for station 1 and 2 should also be included in Figure/Table. | We now present the explicit age results for core 4 in the updated Figure 2 A.

We deliberately chose to not present the age results from cores 1 and 2 because we used the results of Constantinescu et al. (2023) as stated in the method section and the results section. We would like to not show the results of Constantinescu et al. (2023) to prevent any confusion of the reader and to not evoke the impression that these are our own results. |
| Section 3.2: I suggest adding more subsections here. | We agree and have restructured this part. |

| | |
|---|---|
| Line 234 Figure 5: How is the f calculated? Should be included in method section. | We added to the method section 2.5 Data integration and analyses: "The apparent isotopic fractionation factor ($\varepsilon$) was calculated by means of Rayleigh plots (Möbius 2013). From the analysed subset of sediment samples, we used the largest value of the total N content as the reference for the calculation of the remaining fraction (f), which consequently plots at the coordinate origin." |
| Section 3.3: I think this section is more suitable as a part of discussion section. | We disagree because the results of the correlation analysis and the resulting plots have to be presented as results. However, our usage of the word "reconstruct" in the section (L 243, L 245) may indeed signal an interpretation and thus belong to the discussion section. We changed this part to:

"The correlation of Danube DIN load with sediment N content and $\delta15N$ values in the nearshore cores 1 and 2 was examined to develop a simple empirical model for reconstructing historical DIN loads for the period before measurements are available. Using core 2, no meaningful correlation was found (not shown). Using core 1, the DIN load of Danube at Reni station (Kovacs & Zavadsky 2021) correlated significantly with the bulk N content (Pearson's R = 0.80) and less significant with $\delta15N$ (Pearson's R = 0.35, Model 3). The N content-based Models 1 and 2 had average residuals with respect to observed DIN loads of $36 \pm 26$ kt / yr and $42 \pm 31$ kt / yr, respectively (Fig. 6 B, D). The average residuals of the $\delta15N$-based Model 3 were $61 \pm 33$ kt / yr (Fig. 6 F). For the period 1800 – 1950, all three models estimated that the Danube DIN load was 236 to 318 kt / yr in 1800 CE and increased gradually with 0.2 to 0.5 kt / yr2." |
| Section 4.1: This section is particularly hard to follow. The raw data in Figure 3 are hard to connect to these events described here. I think | We think that this is a good idea and added the new Figure 7, which is a timeseries figure with marks for the events we mention in this |

| | |
|---|---|
| at least a time series plot should be presented here (e.g., like fig.3 in Fulton et al., 2012). And the major events/periods should be marked in figures. | discussion section. |
| Line 265: Consider briefly explaining what 'Unit IIb' and other jargons refers to, as not all readers may be familiar with this regional stratigraphic nomenclature. | We agree and have added a brief explanation.

"A sapropel with high organic matter concentration (stratigraphic unit II b, Ross et al. 1970) was deposited in the Black Sea basins during this phase" |
| Line 276: Why did no detect of H2S agree with time frame of zone 2 samples? | Organic matter is remineralised faster in oxic conditions than in anoxic conditions, and the Rayleigh plots in Fig. 5 indicate kinetic fractionation that is characteristic for oxic remineralisation. The absence of a H2S proxy supports this conclusion as the presence of H2S in the photic zone would also imply oxygen-free bottom water. We have added to the new section 4.4:

"The deep ventilation is supported by Cutmore et al. (2024), which have not detected the proxy for H2S in the photic zone during this period (3.9 – 2.7 ka BP) while this proxy (isorenieratene) was always present before and after this period and indicates an exceptional deep ventilation. " |
| Section 4.2: This is a very long section discussing about N sources. It could be broken down into subsection according to time periods. And I am wondering the N loss processes' impacts on N isotopic signals. | We agree and have split this section according to time periods as suggested. With respect to the effect of N-loss processes on the N isotopic signal, we already discuss how oxygen exposure enhances remineralisation (e.g.line 341 ff) and how euxinic conditions preserve biomass and the N-signature within (e.g. line 304 ff). |
| Section 4.3: The section aims to discussion the age offsets in Inorganic $^{14}C$ and has little connection in core topic. I think these contents can be moved to age result section. | We agree and have this section disbanded. Parts relevant for the discussion of N were moved to more suiteable parts of the discussion. We have removed the discussion of TIC entirely to focus the manuscript more on nitrogen. The results from carbonate dating were kept for the case |

| | that these data are relevant for later studies. |
|---|---|
| Typo note: There are a lot of typos. E.g., subscript/superscript typos in lines 130, 197, 206, 210 and so on. | These typos initially escaped our attention and were corrected in the revised manuscript. |

Review 2

| Reviewer's comment | Our response |
|---|---|
| Fig. 1: It would be helpful to include a sub-map that shows the study area within a broader geographical context. | We agree and added a sub map to Figure 1 to make it easier to put the study area into a broader geographical context. |
| Line 194: seems TOC/N ratio is higher than 8 at 35 cm depth, better check it | We agree, this is indeed an error. We will change the sentence to "The molar TOC / N ratio decreased from 15 at the sediment surface to 8 at 18 cm sediment depth, and increased to 10 at 35 cm." |
| Line 200: at station 6, 'the organic carbon and nitrogen content decreases in the upper 3 to 7 cm of the cores', which is not the case from the figure. Better check it. | We agree that this part is not very clear. We have changed it to: "At these stations, the organic carbon and nitrogen content decreases in the upper 3 to 7 cm of the cores (TOC 1.1 to 4.0 %; N 0.10 to 0.42 %, Fig. 3)." |
| Line 203: It's hard to see the trends, better to provided detailed numbers or fitting lines | We would like to not add additional elements such as fitting lines to the already quite condensed Figure 3. However, we have described the trends in more detail in the results part. |
| Line 210: I recommend the authors mark the four zones in Fig. 3, which would be helpful to understand Fig. 4 | We generally agree, but we would like to implement this suggestion a bit different. We improved the explanation of the concept of the d15N vs. N content trends in the introduction, we rewrote the corresponding part of the results, and edited Figure 4 to make the whole concept easier to understand.

We would like to not add additional elements to the already condensed Figure 3, which in our opinion would not help to grasp the concept of |

| | the d15N vs. N content trends, |
|---|---|

**Review 3**

| Reviewer's comment | Our response |
|---|---|
| Furthermore, at least two references are cited in the manuscript but not included in the bibliography (e.g., Stuiver et al. 1998 – line 131, Siani et al. 2000 – line 132) and the formatting of the bibliography is very inconsistent. | These two references are no longer used in the revised manuscript. |
| Beyond the cosmetic issues, I found the manuscript somewhat difficult to follow in that, while there is a stated research aim of identifying natural and anthropogenic nitrogen sources over the past 5000 years, the discussion started with a section of sedimentary signatures of major events in the Black Sea. The abstract also mentioned the difficulty of determining a pristine reference state for nutrient reduction, but this idea was not fully developed in the manuscript. I would recommend breaking down the overarching research aim into smaller sub-aims or sub-questions to help organize the discussion more clearly. | We agree and have restructured the results and discussion sections to increase the readability. |
| I also miss some reference to the novelty of the study. There has already been a lot of work published on nitrogen inputs into the Black Sea and eutrophication of the Danube. Indeed, in the conclusions, the authors acknowledge that many of the results are confirming previous findings. I do think that the finding of early eutrophication of the Danube is novel but that this can be more clearly stated. | We agree, that the novelty of our study needs to be stated more clearly. We have expanded the discussion accordingly. We now state more clearly where our results deviate from the results of Fulton et al. 2012 and Cutmore et al. 2025. |
| Line 30: "This raises the question of what point in time could serve as a realistic reference for nutrient reduction goals, given that the Danube has not had pristine nutrient levels for at least | We agree and will expand the discussion acordingly. |

| | |
|---|---|
| 800 years." - This is an interesting point but is not brought up in the manuscript at all. If defining a pristine reference state is one of the study aims, I think that this should be discussed further. | We added to the discussion:

"If our conclusion is confirmed that the eutrophication of Danube started several centuries before the onset of industrialisation, then this would further imply that Danube was not pristine in the sense of the European Water Frame Directive (WFD) since the Middle Ages. The WFD requests from EU member states to manage water bodies at the river basin level to achieve "good status" for all water bodies, which basically requests a condition with no or minimal human impact (ref) and includes the riverine nutrient loads. However, it is not within the scope of this study to define a pristine reference state for Danube, but our results may guide future studies to collect suitable samples for the reconstruction of historic nutrient loads of Danube." |
| Lines 72-83: Citations needed here. | We added references to Kendall et al. 2007 and to Zhang et al. 2014. |
| Line 131-133: I am assuming by the reference to Stuiver et al. 1998 that the IntCal98 was used for the calibration. However, in Table 2, it seems that for all the calibrated dates, 500 was just subtracted from the 14C age. Was that really the result of the calibration? Additionally, what was the rationale for using IntCal98? As there has been many updates to the curve, I do not think it makes sense not to use the most recent version, i.e., IntCal20. However, it would be more appropriate to use Marine20 as that is specifically for marine sediments.

Furthermore, given that linear interpolation assumes constant sedimentation, using a Bayesian approach, such as that of Bacon, could provide a more robust chronology. Given that the manuscript relies heavily on the chronology, the methods here should be | We agree, the method of 14C dating is not in line with current practice, and we updated the method according to your suggestions. Specifically, we now use the Marine20 calibration curve in conjunction with the calib.org database to determine the ΔR value of the Black Sea shelf. We will use the Calib tool to execute the calibration according to Stuiver, M., and Reimer, P.J., 1993, Radiocarbon, 35, 215-230. All figures, tables, and plots were updated accordingly.

We further agree that the Bacon method (e.g. in Blaauw and Christen, 2011) generally provides more plausible age models with less artifacts such as abrupt changes of the modelled sedimentation rate. However, we have only a very limited number of 14C dates; Core 4 has only 3 data point to support an age model. We thus chose to use simple, step-wise linear interpolation as it does not rely on assumptions, |

| | |
|---|---|
| brought in line with current practice. | which are not supported by the data. |
| Line 207: Were these zones identified solely based on observation or were any statistics used? | The zones were identified on observation without additional statistics. The underlying assumption was, that in periods in which a particular condition dominated the trend in d15N vs [N] is roughly constant. We will clarify this in the method section 2.5. |
| Line 241-243: The description of the model development should be in the methods section. | This part was moved to methods section 2.5. |
| Line 40: Replace "as displayed in" with "as seen" | Done. |
| Line 61: Replace "represents" with "is" | Done. |
| Line 64: Should be Maselli & Trincardi, 2013? | Yes, that was a typo and was corrected. |
| Line 68: Replace "measurements" with "measures" | Done. |
| Line 100: Station 2 depth is 27 m in Table 1 but 22 m here. Which is it? | That's a typo, and was corrected. |
| Line 118: Replace "better" with "less" | Done. |
| Line 127 and later: Italicize scientific names | Done. |
| Line 312: Should be Fulton et al. 2012? | That's a typo, and was corrected. |
| Ensure all references have the same style of formatting. | Done. |

---

## Author Response (AR2)

egusphere-2025-1803, Revision 2
Title: Reconstructing changes in nitrogen input to the Danube-influenced Black Sea Shelf during the Holocene'

**1) General remarks**

We again appreciate the thorough review of our manuscript and the detailed advice from the reviewers. Our detailed response is listed below.

**2) Reviewer comments and our responses.**

Review 1:

| Reviewer's comment | Our response |
|---|---|
| I would recommend some basic grammar editing, as there are a few instances where the wrong preposition or verb form is used. Overall, the writing is still understandable, so this is not a major concern, but it would improve the reading experience. | We have checked grammar and style and edited the text. These minor corrections constitute the majority of the current revisions. |
| At line 19 in the abstract, it states the study spans 6000 years, while in the introduction at line 122, it states 7000 years. | We changed the abstract accordingly. |
| When referring to "Atlantic climate" or "sub-boreal climate," as in lines 30 and 31, I would recommend explicitly noting that these are specific periods during the Holocene with different climate regimes, as simply saying "Atlantic climate" may be misread as a regional climate type rather than a Holocene climate period. For example, you could rephrase to "during the warm and wet Atlantic period" and "during the colder and drier Sub-Boreal period." | We agree and follow this suggestion. |
| Define H2S and DIN at first occurrence, lines 44 and 85, respectively. | We introduced both abbreviations as:

 1) However, hydrogen sulfide ($H_2S$) in euxinic environments reduces the |

| | degradability … |
| --- | --- |
| | 2) … a substantial decrease of the Danube load of dissolved inorganic nitrogen (DIN), which is now below… |
| Please add a citation to lines 240-243 regarding the increased transport of sand from the Sfântu Gheorghe branch. | We added a reference to Constantinescu et al. (2023). |
| Additionally, make sure that Sfântu Gheorghe is spelled consistently here and when it is referenced in the discussion. | We made sure that the Name Sfântu Gheorghe is now spelled consistently throughout the text. |

Review 1:

| Reviewer's comment | Our response |
| --- | --- |
| For figure 6 A, R^2 is 0.98, which is much higher than figure 6 C (R^2 is 0.64) that have similar data distribution. Please check the statistic results. | We thoroughly checked the results from both linear regressions, and the $R^2$ values are stated correctly.

The observation data for Models 1 and 2 are identical, the only difference is the applied linear model. Model 1 is without y-intercept, Model 2 is with y-intercept. The differences in the $R^2$ values are also reflected in slightly higher residuals in Model 2 than in Model 1. |
| And I would suggest make consistent expression with figure 6 (R^2) and text (line 300-301 Pearson's R) | We agree and have changed R in the text to now $R^2$. |